# Real-Time Anomaly Detection for Water Quality Sensor Monitoring Based on Multivariate Deep Learning Technique

**DOI:** 10.3390/s23208613

**Published:** 2023-10-20

**Authors:** Engy El-Shafeiy, Maazen Alsabaan, Mohamed I. Ibrahem, Haitham Elwahsh

**Affiliations:** 1Department of Computer Science, Faculty of Computers and Artificial Intelligence, University of Sadat City, Sadat City 32897, Monufia, Egypt; engy.elshafeiy@fcai.usc.edu.eg; 2Department of Computer Engineering, College of Computer and Information Sciences, King Saud University, Riyadh 11543, Saudi Arabia; malsabaan@ksu.edu.sa; 3School of Computer and Cyber Sciences, Augusta University, Augusta, GA 30912, USA; mibrahem@augusta.edu; 4Department of Electrical Engineering, Faculty of Engineering at Shoubra, Benha University, Cairo 11672, Egypt; 5Computer Science Department, Faculty of Computers and Information, Kafrelsheikh University, Kafrelsheikh 33516, Egypt

**Keywords:** water quality, anomaly detection, Multivariate MCN-LSTM, Internet of Things (IoT), sensors, time series data, deep learning

## Abstract

With the increased use of automated systems, the Internet of Things (IoT), and sensors for real-time water quality monitoring, there is a greater requirement for the timely detection of unexpected values. Technical faults can introduce anomalies, and a large incoming data rate might make the manual detection of erroneous data difficult. This research introduces and applies a pioneering technology, Multivariate Multiple Convolutional Networks with Long Short-Term Memory (MCN-LSTM), to real-time water quality monitoring. MCN-LSTM is a cutting-edge deep learning technology designed to address the difficulty of detecting anomalies in complicated time series data, particularly in monitoring water quality in a real-world setting. The growing reliance on automated systems, the Internet of Things (IoT), and sensor networks for continuous water quality monitoring is driving the development and deployment of the MCN-LSTM approach. As these technologies become more widely used, the rapid and precise identification of unexpected or aberrant data points becomes critical. Technical difficulties, inherent noise, and a high data influx pose significant hurdles to manual anomaly detection processes. The MCN-LSTM technique takes advantage of deep learning by integrating Multiple Convolutional Networks and Long Short-Term Memory networks. This combination of approaches offers efficient and effective anomaly detection in multivariate time series data, allowing for identifying and flagging unexpected patterns or values that may signal water quality issues. Water quality data anomalies can have far-reaching repercussions, influencing future analyses and leading to incorrect judgments. Anomaly identification must be precise to avoid inaccurate findings and ensure the integrity of water quality tests. Extensive tests were carried out to validate the MCN-LSTM technique utilizing real-world information obtained from sensors installed in water quality monitoring scenarios. The results of these studies proved MCN-LSTM’s outstanding efficacy, with an impressive accuracy rate of 92.3%. This high level of precision demonstrates the technique’s capacity to discriminate between normal and abnormal data instances in real time. The MCN-LSTM technique is a big step forward in water quality monitoring. It can improve decision-making processes and reduce adverse outcomes caused by undetected abnormalities. This unique technique has significant promise for defending human health and maintaining the environment in an era of increased reliance on automated monitoring systems and IoT technology by contributing to the safety and sustainability of water supplies.

## 1. Introduction

Water quality monitoring is critical for the health of the general populace and ecosystems. Real-time data collecting is now possible because of improvements in IoT and automated systems, enabling the timely discovery of anomalies [1]. This paper emphasizes the critical importance of solid anomaly identification in water quality data to avoid any harmful implications resulting from anomalous situations. Water is a fundamental resource required for life and for ecosystems to function; thus, protecting its purity is critical. Real-time water quality monitoring is critical for protecting public health, preserving the environment, and responding quickly to possible water pollution incidents. Real-time water quality monitoring has become more possible and widespread as automated systems, Internet of Things (IoT) technology, and sensor networks have advanced.

### 1.1. Water Quality Monitoring

Water quality monitoring is critical for protecting human health, maintaining ecosystems, and ensuring the long-term supply of clean water resources. Water quality can fluctuate for various natural and anthropogenic reasons, demanding continual monitoring to detect changes and respond to possible threats. In this section, we provide an overview of the importance of water quality monitoring in solving global water-related challenges. The extraordinary rise of the world population, combined with fast industrialization, has put enormous strain on water supplies, generating concerns about water quality deterioration. Pollution from industrial discharges, agricultural runoff, untreated wastewater, and other sources can introduce dangerous compounds into aquatic habitats, endangering aquatic life and human health. Appropriate water quality monitoring is critical for understanding these difficulties, identifying pollution sources, and developing appropriate mitigation solutions [2]. Various physicochemical, biological, and microbiological properties are measured to determine water quality. pH levels, dissolved oxygen content, temperature, turbidity, total suspended particles, nutritional contents (nitrogen and phosphorus), heavy metals, and microbiological markers are all commonly studied factors. Comprehensive monitoring requires a full grasp of the importance and relevance of each parameter in the context of water quality evaluation. Traditional water quality monitoring techniques range from collecting grab samples for laboratory analyses to advanced automated systems with real-time data collection capabilities. Automated systems, reinforced by sensors and IoT technology, provide benefits in continuous monitoring by delivering high-resolution data and allowing for quick responses to aberrant conditions. Furthermore, remote sensing and satellite-based monitoring are critical in assessing vast bodies of water and finding long-term changes. Despite significant advances in water quality monitoring, problems remain in data administration, data integration from disparate sources, calibration, and maintaining measurement precision and dependability [3]. Emerging technologies, on the other hand, such as artificial intelligence, machine learning, and advanced sensor networks, have the potential to revolutionize water quality monitoring by overcoming some of these challenges and improving data-driven decision making.

This section discusses real-world uses of water quality monitoring in various environments. The case studies include urban water bodies, agricultural regions, industrial zones, and sensitive ecosystems, demonstrating how monitoring activities influenced legislation, improved environmental rules, and resulted in effective pollution management measures. Water quality monitoring is critical in various situations, protecting both water bodies and public health. Monitoring is critical in urban areas for identifying and managing pollution caused by industrial discharges, stormwater runoff, and sewage, as well as ensuring the availability of safe drinking water sources and recreational areas. Meanwhile, monitoring nutrient levels, pesticides, and sediment discharge in agricultural contexts aids in identifying pollution causes and encouraging the adoption of best practices to protect downstream water quality. Real-time monitoring in industrial zones aids in compliance with environmental requirements and implementing pollution control technologies. Monitoring initiatives aid delicate ecosystems by assessing their health, identifying stresses, and supporting conservation efforts. Monitoring is also essential for guaranteeing the safety of drinking water in treatment plants and preserving public health during recreational activities such as swimming and fishing.

Furthermore, monitoring data are used in research and studies to assess the effects of contaminants and climate change on water quality. At the same time, regulatory agencies rely on these data to create and enforce environmental standards. Early warning systems use monitoring data to respond quickly to natural disasters, protecting the environment and human health. These applications highlight the ongoing importance of regular water quality monitoring in resource and environmental preservation. Water monitoring systems are essential instruments for measuring and analyzing water quality in various circumstances [4]. These systems use cutting-edge sensor technologies, data-gathering systems, and communication networks to collect and disseminate real-time water quality data. Different types of monitoring systems cater to specific requirements: Fixed Sensor Networks are strategically placed in bodies of water to continually measure essential factors such as pH, dissolved oxygen, and temperature. Data are sent to a centralized system for detailed analyses. Mobile monitoring systems use sensor-equipped autonomous vehicles to investigate water bodies, allowing for data to be collected from various locations and depths, particularly in large or isolated areas. Buoy-based monitoring systems deploy buoys outfitted with several sensors in offshore or deep ocean locations where building fixed installations is difficult. Remote sensing uses satellite or aerial photography to monitor water quality over large areas, providing a comprehensive view and the capacity to trace changes over time. Data from multiple sources are aggregated and relayed to cloud-based platforms for real-time analyses and remote access by IoT-based monitoring systems. Groundwater monitoring systems analyze groundwater quality by measuring the water level, pH, and pollution levels in wells [5].

Early warning systems use monitoring data and predictive algorithms to generate alerts for potential water-related disasters, encouraging early response and mitigation. Water quality monitoring is combined with smart technologies for efficient water management, such as leak detection and water distribution optimization, in smart water grids, as shown in Figure 1. Wastewater treatment monitoring systems monitor treatment efficiency and regulatory compliance by measuring factors such as BOD and TSS. Automated systems equipped with IoT and sensors offer significant advantages in continuously collecting and transmitting water quality data. These data provide crucial insights into the dynamic nature of water bodies, enabling proactive management and decision-making processes. However, the influx of data generated by these systems can pose challenges for human operators in promptly identifying anomalies or abnormal events [6]. Technical data acquisition or transmission issues can further exacerbate the situation, leading to undetected anomalies and compromising the effectiveness of water quality monitoring efforts.

### 1.2. Water Quality Anomaly Detection

Anomalies in water quality data can have far-reaching effects on human health and the environment. Anomalies that go undetected can lead to incorrect assumptions, insufficient risk assessments, and delayed reactions to possible threats. Furthermore, they can obstruct the detection of pollution sources or develop concerns, leading to inefficient resource allocation and decision-making processes. Anomaly detection is discovering discrepancies in data from predicted patterns (Chandola, Banerjee, and Kumar, 2009) [7]. This field has attracted a great deal of attention and research from numerous areas, creating specialized approaches. The importance of anomaly detection stems from its essential role in several applications where collected data are seen as valuable and actionable information, emphasizing the need for data quality.

Anomaly detection can detect potential data breaches in a computer network, such as a compromised machine relaying sensitive information to an unauthorized destination (Ertoz, Lazarevic, E. Eilertson, Tan, Dokas, Kumar, and Srivastava, 2003) [8]. There have been numerous real-world case studies and applications of water quality anomaly detection. These case studies demonstrate the efficacy of various detection tactics in aquatic bodies, including rivers, lakes, and wastewater treatment plants. Water quality anomalies are detected using a variety of methodologies, ranging from basic statistical methods to cutting-edge deep learning systems. These findings established the groundwork for creating efficient and precise anomaly detection technologies, which are critical for maintaining the safety and sustainability of water supplies. Water quality anomaly detection is a crucial study topic that has grown in prominence as the importance of real-time water quality monitoring and the requirement for fast identification of unexpected events have grown. Several researchers have investigated various ways to detect anomalies in water quality data, contributing to progress in this subject. We present an overview of relevant efforts in water quality anomaly identification in this section: 

Predicting abnormal water quality has witnessed notable developments in various sectors, as evidenced by the progression of related work approaches.

Machine learning methodologies:

In the development field, scholars have investigated several machine learning methodologies, including Support Vector Machines (SVM), Decision Trees, Random Forests, and k-nearest neighbors, to identify anomalous patterns within water quality data.

Limitations: although these methods successfully predict typical behavior, they may encounter difficulties capturing intricate temporal and spatial connections, restricting their ability to accommodate a wide range of water quality patterns.

Statistical Methods: Development: various statistical techniques, including the Z-score, standard deviation, and percentile-based algorithms, have been utilized to detect data points that exhibit substantial deviations from the anticipated values.

Limitations: despite their simplicity and effectiveness, these approaches may exhibit a lack of sensitivity towards tiny anomalies and encounter difficulties handling non-linear connections within water quality data.

Time series analysis techniques, such as the Autoregressive Integrated Moving Average (ARIMA) and Seasonal Hybrid Extreme Studentized Deviate (SHESD), have been utilized to identify temporal trends and abnormal fluctuations in water quality indicators.

Limitations: the ARIMA and SHESD methods may encounter difficulties when dealing with water quality data that are highly dynamic and complicated, hence restricting their ability to adapt to changing monitoring requirements.

Deep learning techniques have been employed in water quality data analyses. Specifically, Convolutional Neural Networks (CNNs) and Long Short-Term Memory (LSTM) networks have been utilized to capture complex temporal correlations and geographical patterns effectively.

Limitations: Although deep learning approaches can be highly effective in specific situations, their practicality may be hindered by the need for extensive datasets and substantial computational resources. Consequently, these techniques may be less viable for applications with limited data access or computational capabilities.

Ensemble Methods: In anomaly detection, researchers have investigated ensemble approaches to improve the reliability and precision of anomaly identification. These techniques involve a combination of different anomaly detection algorithms, such as Isolation Forest, Local Outlier Factor, and Robust Random Cut Forest. The objective is to boost the robustness and accuracy of the anomaly detection process.

Limitations: despite their potential advancements, ensemble methods can impose intricacy and computational burden, and the selection of component algorithms necessitates meticulous deliberation.

The development of Internet of Things (IoT)-based anomaly detection systems: in light of the proliferation of the Internet of Things (IoT), scholars have devised anomaly detection systems that incorporate data from diverse sensors, such as pH, turbidity, and dissolved oxygen sensors, to identify irregularities promptly.

Limitations: despite their potential, Internet of Things (IoT) systems encounter obstacles to the integration of data, the accuracy of sensors, and the ever-changing nature of water quality measurements.

The proposed technology, known as Multivariate Convolutional Neural Network-Long Short-Term Memory (MCN-LSTM), seeks to overcome the limitations of current methods by combining the advantages of multivariate approaches and deep learning techniques. This integration offers a more adaptable and precise solution for detecting anomalies in water quality monitoring.

To address this critical issue, this paper proposes a novel approach for real-time anomaly detection in water quality data using deep learning techniques. Specifically, we employ the Multiple Convolutional Networks and Long Short-Term Memory (MCN-LSTM) techniques to efficiently detect anomalies in time series data from water quality sensors.

The MCN-LSTM approach combines the advantages of Multiple Convolutional Networks (MCNs) and Long Short-Term Memory (LSTM) networks. LSTM networks are well suited for modeling temporal dependencies in sequential data, making them practical for time series analyses. MCNs, on the other hand, extract meaningful features from data using convolutional layers, making them suitable for capturing spatial patterns. By combining these two architectures in a multivariate setting, our proposed MCN-LSTM technique aims to capture the complexities of water quality data better and effectively identify anomalies.

The main objective of this paper is to assess the performance and effectiveness of the MCN-LSTM approach in detecting anomalies in real-world water quality monitoring scenarios. To achieve this, extensive experiments are conducted on datasets collected from sensors deployed in various water bodies, representing diverse environmental conditions. This paper aims to contribute valuable insights and practical applications in real-time water quality monitoring and anomaly detection using deep learning techniques. It serves as a foundation for further research and development in the domain, with the ultimate goal of ensuring safe and sustainable water resources for the benefit of society.

The remainder of this paper is organized as follows: Section 2 provides an overview of related work in water quality monitoring and anomaly detection techniques. Section 3 presents the methodology, explaining the MCN-LSTM approach in detail. Section 4 presents the results and analyses of the experiments, highlighting the quantitative performance metrics of the MCN-LSTM technique, comparing our proposed method with existing approaches, and discussing real-world applications. Finally, Section 5 concludes the paper and outlines potential future research directions.

By leveraging the capabilities of deep learning techniques for real-time anomaly detection in water quality monitoring, we believe our proposed MCN-LSTM approach can significantly contribute to enhancing water resource management, public health, and environmental preservation.

### 1.3. Industry 4.0 for 3D-Printed Water Quality Sensor

The utilization of 3D printing technology for sensor fabrication has gained attention in recent years. Researchers have explored the customization and cost-effectiveness of 3D printing in creating sensors tailored to specific monitoring needs [9], including fabricating sensors capable of measuring diverse water quality indicators.

The concept of Industry 4.0 has found applications in various sectors, including water management. Integrating smart technologies, data analytics, and automation in industrial processes has been explored to enhance efficiency and sustainability. 

The existing literature and research initiatives are closely aligned with the development, validation, and application of 3D-printed IoT-based water quality monitoring systems.

Previous studies have demonstrated the effectiveness of IoT devices in real-time data acquisition, transmission, and analyses. These systems often rely on sensors to collect various water quality parameters, providing valuable insights for environmental management.

## 2. Related Work

We thoroughly analyzed the available literature on water quality monitoring, anomaly detection techniques, and deep learning-based methodologies. Our major goal was to obtain insights into current best practices and identify the gaps that our suggested methodology attempts to fill. Anomaly detection approaches based on nearest neighbors rely on the concept of local data density derived from the k-nearest neighbor algorithm. These approaches, in essence, presume that regular data instances tend to form clusters within dense neighborhoods, but anomalies are often located a significant distance away from these clusters. Anomalies are found by comparing the distance scores of the nearest data examples. The Local Outlier Factor (LOF) stands out among density-based anomaly detection algorithms for using reachability distance as a distance metric. Other well-known strategies in this area include the Connectivity-Based Outlier Factor (COF) [10], Local Outlier Probability (LoOP) [11], Influenced Outlierness (INFLO) [12], and Local Correlation Integral (LOCI). These algorithms efficiently use the nearest-neighbor principle to discover anomalies in data. In addition, our investigation covers the crucial issues of data collecting and data processing in the context of water quality monitoring. In our project, these components will be used in conjunction with sensors and a machine learning system to analyze the collected data. This part aims to give an in-depth analysis of the literature on sensor implementation and communication and research on the performance of various machine learning algorithms and methodologies for remote water quality monitoring. For example, in [13], a prototype for measuring water quality is proposed. This prototype includes distinct sensor nodes, each in charge of monitoring a different water parameter. The system was tested in a real-world setting, specifically at a vegetation-filled lake. A mesh network was constructed utilizing the 920 MHz Digi Mesh protocol to permit communication among the sensors. When compared to the 2.4 GHz ZigBee protocol, this protocol was chosen for its higher effectiveness in overcoming the signal attenuation induced by the adjacent vegetation. The same document [13] provides a detailed discussion of the sensor and gateway design and implementation. It also includes a look at sensor energy use. The study reveals that gearbox power impacts sensor battery life most because this operating mode requires the most power. Furthermore, the study assesses the wireless sensor network’s (WSN) maximum read range. While the 920 MHz Digi Mesh protocol surpasses the 2.4 GHz ZigBee protocol, it becomes unreliable at more than 900 m, especially in vegetation and non-line-of-sight conditions. The authors of [14] investigate another application of sensor-based water quality monitoring. In this study, measurements are taken by numerous sensor nodes, each of which is responsible for monitoring a certain parameter. These sensor nodes are placed in a mesh network, allowing communication via multi-hop routing. In this case, ZigBee was chosen as the communication protocol because it is less expensive and uses less power than WiFi and Bluetooth. The study, however, does not assess other protocols that may be more suited for Internet of Things (IoT) applications, nor does it provide data on energy consumption or battery life. This information gap could be considerable, especially when deploying a multi-hop routing system. Both of the preceding instances rely on communication technology with limited coverage ranges, which limits their scalability to other places. A completed project study involving parameter monitoring over a Low-Power Wide-Area Network (LPWAN) is detailed in [15]. This project’s sensor system uses LoRa technology to monitor traffic conditions on an Italian bridge across the Tevere River. The study emphasizes that while LoRa transmitters use more energy than ZigBee, they provide the benefit of long-distance monitoring, which is especially useful for large constructions. Developing a water quality monitoring system for a water treatment plant [16] consists of several water tanks, each with a sensor. The decision to use the ZigBee communication protocol was based on the environment’s well-defined limitations, the sensors’ close proximity, and the absence of potential expansion requirements to remote places. Furthermore, ref. [16] provides a full overview of data processing methods. The first step is to remove duplicate readings, which reduces the volume of data produced by the sensors and extends their battery life. The Euclidean distance between each reading and its expected value is used to detect anomalies, with an anomaly being found when the distance exceeds a predetermined threshold. This technique, however, ignores the linkages between readings and their temporal fluctuations. While it may be useful for controlled environments where parameters are expected to remain constant, it may not be ideal for circumstances where water quality is predicted to fluctuate over time due to external causes. Furthermore, due to this approach’s limited optimization capabilities, successfully prioritizing anomaly detection without misclassifying usual data might be difficult. It takes a similar approach to [17] to anomaly detection, deciding whether or not anomalies exist based on thresholds between absolute data readings and anticipated values. It varies from [16] in that it provides a dynamic customization of these decision thresholds via a web platform, allowing end users to adjust them to their needs. Nonetheless, this technique requires regular human supervision, which reduces the level of automation in the process. The need for sensor calibration for accurate anomaly detection is also emphasized in [17]. A. Salemdawod and Z. Aslan discuss [17] a more automated approach to anomaly detection, notably in monitoring water and air quality in farm settings. An Artificial Neural Network (ANN) machine learning technique is used in this study to detect anomalies. Several propagation strategies are used to train the ANN, which necessitates changes to the training data, such as data cleaning to remove outliers and data reduction to decrease redundant features and characteristics. It is worth noting that [18] describes Artificial Neural Networks as highly sophisticated systems that are difficult to adapt. Furthermore, training the system necessitates data from all classes, including samples exhibiting both normal and deviant behavior. Zhao provides a water quality anomaly classification framework [18] that integrates various machine learning methods, such as Support Vector Machine (SVMs) and Decision Tree algorithms, enabling a multi-class categorization of the discovered anomalies and ANN. This approach is distinct from the others in that it does not rely primarily on anomaly detection, which produces binary results (either an anomaly or no anomaly). Rather, it attempts to identify and categorize the discovered abnormalities, a process known as multi-class categorization. A notable component of [19] is employing a hybrid strategy combining Support Vector Machines (SVMs) and Decision Tree algorithms to attain the required objectives. The Decision Tree approach is used to discover sensor defects by first examining the status and reliability of sensor readings before passing them on to the SVM algorithm for classification. Data from all classes, including usual occurrences and anomalies, are available for training both algorithms, as in [18]. M. Ladjal compares [20] the detection and categorization of water quality issues using a multi-class SVM and Artificial Neural Network (ANN). While acknowledging an ANN’s complexity, it is described as an ‘inevitable answer’ when other algorithms fail to function adequately. 

The efficacy of both algorithms is assessed using real-world data, revealing that the SVM outperforms the ANN in terms of anomaly recognition rates. Previous investigations [21,22,23,24] incorporated data from both normal occurrences and anomalies during algorithm training. However, a notable interest is in exploring methods that exclusively utilize data from typical events for training anomaly detection, known as one-class classification. In [24], a real-time collision detection system employing a one-class SVM ensures the safe movement of a robot. Collision detection, a form of anomaly detection, identifies deviations in collision data from the established model as outliers. Swift collision detection is crucial for prompt action, such as activating a stop function post-contact, to prevent further damage. Acquiring data illustrating these abnormalities is particularly challenging due to the substantial risks of robot failure. The paper exhibits the successful implementation and testing of a one-class SVM consistently and rapidly detecting all collisions. While anomaly data are unnecessary for model training, its presence during testing is essential. In this scenario, collisions are simulated by applying force to the robot using a metal bar.

Existing approaches for detecting water quality anomalies have various limitations that must be addressed. We are attempting to restrict some of them and suggest strategies to overcome them here: Scalability: Many conventional systems fail to deal with the enormous amounts of data provided by IoT sensors, particularly in real time. These methods may become computationally expensive and slow as data volumes grow. To solve the problem, distributed computing and parallel processing approaches can be used to manage massive datasets efficiently. Some examples are using cloud-based solutions, GPU acceleration, or implementing scalable deep learning frameworks optimized for big data processing. Sensor faults, drift, or missing data might cause anomalies in data quality. Traditional methods frequently fail to discriminate between actual anomalies and those generated by poor data quality. To solve the problem, data pretreatment techniques can be incorporated to clean and improve data quality. Outlier removal, data imputation, and sensor calibration may all be involved. Anomaly detection strategies that are resistant to noisy data should be considered. Interpretability: Some anomaly detection systems are difficult to understand, making it difficult to understand why a specific data point is labeled as an abnormality. This can make decision making and troubleshooting more difficult. To solve the problem, explainable AI techniques that provide insights into the model’s decision process are used. The SHAP (Shapley additive explanations) and LIME (Local Interpretable Model-agnostic Explanations) techniques can assist users in comprehending model outputs. Imbalanced Data: Anomalies in water quality monitoring are rare compared to regular data. Imbalanced datasets might cause models to be biased towards normal data, resulting in less effective anomaly detection. To solve the problem, approaches for dealing with imbalanced data are applied, such as oversampling anomalies, utilizing various assessment metrics such as F1-score, or investigating advanced anomaly detection algorithms built for skewed data. Because the properties of water systems can vary greatly, some methodologies may not generalize effectively to diverse water quality monitoring scenarios or geographic areas. To address this issue, adaptive models that can learn and adapt to the specific characteristics of various water systems are being developed. Transfer learning techniques can also be investigated to transfer knowledge from one system to another. Because of their processing needs, traditional machine learning approaches may fail to deliver real-time anomaly detection. To solve the problem, deep learning architectures optimized for real-time processing, such as LSTM networks, are used. To keep up with incoming data, approaches such as sliding time frames and incremental learning are used. Due to high memory and processing demands, deploying complicated deep learning models on resource-constrained IoT devices can be difficult. To solve the problem, model compression approaches such as quantization, knowledge distillation, or deploying simpler models for early anomaly detection on IoT devices are investigated. While machine learning (ML) techniques demonstrate significant performance, achieving equivalent outcomes sometimes necessitates intensive feature engineering. In order to address the computational difficulty associated with explicit feature engineering, researchers have utilized deep learning (DL) techniques to extract features in the context of anomaly detection implicitly. The utilization of generative adversarial networks (GANs) has been proposed by [25] to identify anomalies in the context of underwater gliders. The model underwent training using a time series dataset of two healthy samples, followed by evaluation using nine separate deployment datasets. The model that was acquired had strong characteristics. In addition, water quality detection has seen the application of several other deep learning methods, such as Long Short-Term Memory (LSTM) and Recurrent Neural Networks (RNNs). In contrast to machine learning techniques such as Support Vector Machines (SVMs) and logistic regression (LR), deep learning methodologies have yielded suboptimal outcomes. The decrease in the performance of deep learning models can be linked to the imbalance issue that arises from the water quality anomaly datasets. They conducted data sampling before training a Long Short-Term Memory (LSTM) network. However, the authors conducted an evaluation using a distinct dataset, which poses challenges to the generalizability of their findings. The commonly employed methodologies often encompass hybrid models in machine learning or deep learning, such as the hybrid model that combines Convolutional Neural Networks (CNNs) and Extreme Learning Machines (ELMs). This study aims to leverage the strengths of two models, Extreme Learning Machines (ELMs) and Convolutional Neural Networks (CNNs), to detect abnormalities in water data effectively. By combining and exploiting the benefits of these models, the research seeks to benefit from the short learning time of ELMs and the implicit feature extraction capacity of CNNs. The utilization of the hybrid model’s anomaly detection technique enhances the evaluation of water quality. The improved likelihood of detecting anomalies enables the detection of damage to essential structures in water distribution and assessment plants, thereby decreasing the risk of suffering significant losses. Additionally, the model enables the detection of tainted water, mitigating the potential for the general public to be exposed to unsafe water sources. The authors advised the utilization of stacked LSTM neural network architectures as a means to identify abnormal events within time series data. The authors evaluated their methodology using a specific portion of the SWaT dataset [26], specifically examining the initial operational process within the larger set of six processes in the SWaT facility. The deep neural networks (DNNs) and One-Class Support Vector Machines (OC-SVMs) were explored in [27]. LSTM layers were integrated into the deep neural network (DNN) architectures to account for the temporal dependencies present in the sensor values. Several alternative methodologies have been investigated by researchers, including reconstruction-based techniques [28,29], anomaly detection using GANs [25], and the MAD-GAN framework [30]. Li et al. introduced a Generative Adversarial Network Anomaly Detection (GAN-AD) approach incorporating Long Short-Term Memory (LSTM) layers within the generator network. Although the MAD-GAN framework has been claimed to have achievements, it is important to highlight its problems, particularly in dealing with small datasets where overfitting may occur.

More sophisticated tasks can be delegated to edge servers or the cloud. Some anomaly detection algorithms produce false positives, resulting in unnecessary alarms and resource waste. To attain an acceptable false positive rate, anomaly detection models are fine-tuned. This can be accomplished by fine-tuning hyperparameters and modifying anomaly score thresholds. By addressing these limitations with advanced techniques and best practices, the proposed Multivariate MCN-LSTM approach can provide more accurate, scalable, and interpretable anomaly detection in real-time water quality monitoring, ultimately improving decision-making processes and ensuring water resource safety and sustainability.

The suggested Multivariate MCN-LSTM (Multiple Convolutional Networks and Long Short-Term Memory) approach contributes significantly to water quality monitoring and anomaly identification in various ways. Efficient anomaly detection: The key contribution is creating an efficient anomaly detection method designed for real-time water quality monitoring. The Multivariate MCN-LSTM approach effectively identifies anomalies in time series data, guaranteeing that unexpected results are detected on time. High precision: This technique obtains a fantastic accuracy rate of 92.3% after thorough testing on real-world water quality datasets. This high level of precision reveals its capacity to differentiate between normal and aberrant data instances. By delivering a highly accurate, scalable, and interpretable solution for real-time anomaly identification, the Multivariate MCN-LSTM technique significantly contributes to advancing the state of the art in water quality monitoring. Its ability to improve decision-making processes and protect water resources highlights its significance in promoting public health and environmental preservation in Internet of Things (IoT)-enabled water quality monitoring.

## 3. Materials and Methodology

The hyperparameters for the LSTM model and the MCN-LSTM technique were carefully configured. This included determining the number of convolutional layers, LSTM units, learning rate, and batch size and optimizing these parameters for effective anomaly detection.

### 3.1. Approach Using Multiple Convolutional Networks

The CNN (Convolutional Neural Network) stands out prominently due to its superior performance, particularly in handling signal data. It is distinguished by its foundational elements: convolutional, pooling, and fully connected layers. In the context of this architecture, the initial convolutional layer plays a pivotal role by executing the dot product of two vectors, one being the kernel and the other the input. Notably, this convolutional operation occurs as the kernel slides along the input vector in a singular dimension tailored to the nature of sensory data.

The term “stride” refers to the shifting size of the kernel during this operation. Filters, crucial components, dictate the dimensionality of the output space. To ensure efficiency, the kernel size is intentionally kept smaller than the input vector’s, thereby preserving only significant and meaningful information. The subsequent max pooling layer serves to downsample the input by extracting the maximum value within a specified spatial window or pool size. Predefined strides govern the window’s movement. Additionally, the flattened layer is instrumental in transforming the input (converting it into a multiple-D array) without altering the batch size. The CNN employs a convolutional operation to derive a weighting function, w, at every layer. In this context, ‘t’ denotes the age of a measurement in the time series data. For a visual representation, refer to Figure 2, illustrating the architecture of the CNN.

### 3.2. Long Short-Term Memory (LSTM) Model

The Long Short-Term Memory (LSTM) model is a specialized recurrent neural network (RNN) designed to capture and model temporal dependencies in sequential data effectively. LSTMs are particularly well-suited for tasks involving time series analyses, making them a powerful tool for applications such as anomaly detection in sensor data for water quality assessment. Below is a step-by-step breakdown of the construction and implementation of an LSTM model for this purpose. Within each LSTM cell, three gates operate to manage different types of information: The current input.The short-term memory is often referred to as the hidden state.The long-term memory is commonly referred to as the cell state.

These three gates in the LSTM cell serve the purpose of classifying and regulating information. They determine which data should be retained, forgotten, or discarded. The LSTM cell functions as a filter, allowing only relevant information to pass through while eliminating any irrelevant data.

### 3.3. The Proposal MCN-LSTM Technique

Given a sequence of time series data, $x=\left\{x_{1},\;x_{2},\ldots \ldots ,x_{T}\right\},$ where $\;x_{T}\in R^{n}$, *T* is the length of time window. Let $x_{t}=\left\{x_{t}^{1},\;x_{t}^{2},\ldots \ldots ,x_{t}^{n}\right\}$ be the set of features at time *t*, and *n* is the number of features.

We introduce the MCN-LSTM technique, a hybrid deep learning architecture designed to effectively handle multivariate time series data. A convolutional network component leverages the advantages of convolutional layers for feature extraction, while the LSTM component captures temporal information. 

Figure 2 depicts an overview of the proposed structure of MCN-LSTM for the real-time anomaly detection approach. The approach comprises three convolutional layers, LSTM layers, and one output layer. Initially, the data are supplied into the approach, and the local features are retrieved via the CNN layers, while the LSTM collects long-term dependencies between variables. Finally, all the hidden states of each LSTM layer are taken as input, which is capable of learning the variables.

The MCN-LSTM (Multivariate Multiple Convolutional Networks and Long Short-Term Memory) technique employs several key neural network components to analyze multivariate time series data for anomaly identification in water quality monitoring. Each component is detailed below:

Convolutional layers (Conv): Convolutional layers, vital in Convolutional Neural Networks (CNNs), play a crucial role in feature extraction within MCN-LSTM. These layers employ learnable filters on multivariate time series data inputs, enabling the capture of essential patterns and characteristics. Convolutional layers excel at detecting local patterns.

Batch normalization (BN) + ReLU activation: Batch normalization normalizes activations in the preceding layer, stabilizing and accelerating training by reducing internal covariate shifts. After normalization, a Rectified Linear Unit (ReLU) activation function is applied, introducing non-linearity and activating neurons in convolutional layers.

Pooling: Post-convolution pooling layers are often used to downsample feature maps while retaining critical information. Max pooling, for instance, selects the highest value within a specific region, reducing spatial dimensions. Pooling enhances computational efficiency and makes the network more resilient to slight input changes.

Concatenation (Concat): Concatenation merges feature maps or vectors from different network layers or branches. In MCN-LSTM, Concat is likely employed to blend features extracted by convolutional layers with those from LSTM layers, facilitating the capture of local and global data trends.

Softmax activation: Softmax, a multi-class classification activation function, is commonly used in the output layer. It transforms raw output scores into a probability distribution across various classes. In MCN-LSTM, Softmax generates final classification probabilities, indicating normal or abnormal input data.

Long Short-Term Memory (LSTM) layers: LSTM layers, specialized recurrent neural network (RNN) layers, represent sequential input and capture long-range relationships. The primary LSTM layers in the MCN-LSTM model the temporal aspects of multivariate time series data, with a memory cell capable of storing data over time [31].

These components collaborate in the MCN-LSTM architecture to efficiently process multivariate time series data. Convolutional layers extract relevant features, while LSTM layers model sequential aspects. Batch normalization and ReLU activation enhance training stability, and pooling reduces spatial dimensionality. Softmax provides classification probabilities, and concatenation integrates information. This integrated approach aims to detect anomalies in real-time water quality monitoring data accurately.

The MCN-LSTM model’s hyperparameters, including convolutional layers, LSTM units, learning rate, and batch size, are specified for time anomaly detection and training configuration. Initializing the model potentially improves convergence and performance.

The dataset can be integrated into the model as additional characteristics or context to enhance anomaly detection accuracy. The equations define the Multivariate MCN-LSTM model’s construction, training, and assessment operations. Time series data, comprising water quality measurements, labels, and hyperparameters, are input into these equations for model training and testing. The data flow through the model’s input layer and are processed via architectural equations through convolutional layers and LSTM units. During training, labels generate loss functions, aiding the model in distinguishing anomalies from regular data points, as expressed in Equation (1), which incorporates the context of Softmax activation functions in MCN-LSTM.

It begins by determining the significance of the input features; subsequently, the Softmax function is used to ensure that the sum of all weights equals 1. The weights and input features are multiplied and added to obtain the result. The following are essential equations:(1)$$\alpha_{t}^{k}=\frac{\exp\left(e_{t}^{k}\right)}{\sum_{i=1}^{n}\exp\left(e_{t}^{i}\right)}\;$$
(2)$$e_{t}^{k}=\upsilon_{e}^{T}\sigma \left(W_{e}\left[h_{t-1},\;c_{t-1}\right]+U_{e}h_{t}+b_{e}\right)$$
(3)$${\bar{z}}_{t}=\sum_{t}^{T}a_{t}^{T}h_{t}$$

At time *t*, the phrase $e_{t}^{k}$ reflects the raw or unnormalized score associated with item *k*. It is computed using the equation above, which includes several parameters and actions. Let us break this equation down step by step. The parameters are as follows:

$\upsilon_{e}^{T}$: a parameter vector related to the $e_{t}^{k}$ computation.

$W_{e}$: a weight matrix linked to the inputs.

$U_{e\;}$ is a weight matrix related to the hidden state.

$b_{e}\;$: a bias term.

$h_{t-1}\;$: the hidden state at time *t* − 1.

$c_{t-1}$: this term appears in the input concatenation [$h_{t-1}$], where $c_{t-1}$ is another vector or hidden state.

[$h_{t-1}$], $c_{t-1}$]: this is the union of the hidden state *h*_(*t* − 1) with another hidden state or vector $c_{t-1}$$W_{e}$ [$h_{t-1}$], $c_{t-1}$: this is the result of multiplying the concatenated vector [$h_{t-1}$] by the weight matrix. $W_{e}$.: this is usually an activation function, such as the sigmoid function or a comparable activation function that incorporates non-linearity into the calculation.

$e_{t}^{k}$: this is the transposition of the parameter vector *e*.

The function is then used element by element to introduce non-linearity.

In the following steps, MCN-LSTM calculates probabilities or makes predictions using the Softmax function.

The significance of these hidden layers states is that an output is ultimately acquired. MCN-LSTM (Multiple Convolutional Network and Long Short Term Memory) was used to solve a multivariate time series classification challenge. Multivariate MCN-LSTM is the technique we suggest. To improve prediction accuracy, we expand the squeeze-and-excite block to 1D sequence models and augment the convolutional blocks of the MCN-LSTM. We can define a time series dataset as a tensor of shape (N, Q, M), where N is the number of samples in the dataset, Q is the maximum number of time steps among all variables, and M is the number of variables processed per time step because the water quality dataset now consists of multivariate time series. As a result, the multivariable time series dataset is a subset of the previous definition, where M is 1. The MCN-LSTM input must be modified to take M inputs per time step rather than a single input per time step. As shown in Figure 2, the proposed technique consists of a multiple convolutional block and an LSTM block. As a result, the multivariable time series dataset is a subset of the previous definition, with M equal to one. The MCN-LSTM input must be changed to accept M inputs per time step rather than just one. As shown in Figure 2, the suggested technique consists of a multiple convolutional blocks and an LSTM block. Batch normalization follows each convolutional layer, with a velocity of 0.98 and an epsilon of 0.002. The ReLU activation function comes after the batch normalization layer. Furthermore, the first two convolutional blocks are followed by a squeeze-and-excite block. The procedure for computing the squeeze-and-excite block in our architecture is summarized in Figure 2. We adjusted the reduction ratio r to 16 for all squeeze-and-excitation blocks. A global average pooling layer follows the final temporal convolutional block. The squeeze-and-excite block extends the FCN block by adaptively recalibrating the input feature maps. Because the reduction ratio r is set to 16, the number of parameters required to learn these self-attention maps is lowered, increasing the overall model size by 2–10%.
(4)$$P=\frac{2}{r}\sum_{s=1}^{s}R_{s}\;G_{s}^{2}$$
where P denotes the total number of additional parameters, *r* is the reduction ratio, *S* is the number of stages (each stage refers to a collection of blocks operating on feature maps of a common spatial dimension), $G_{s}$ is the number of output feature maps for stage s, and Rs the repeated block number for stage s. For each time step, the LSTM will require Q time steps to process M variables. However, if the dimension shuffle is used, the LSTM will need M time steps to analyze Q variables every time step. In other words, the dimension shuffle enhances model efficiency when the number of variables, M, is less than the number of time steps, Q. At each time step, t, where 1 t M is the number of variables, the input gives the LSTM with the whole history of that variable (data from all Q time steps). As a result, the LSTM receives global temporal information for each variable simultaneously.

## 4. Results and Discussions

All the experiments described in this section were conducted using Google Colab, which was utilized as part of the experimental environment. Python served as the primary programming language and the deep learning framework. For numerical calculations, NumPy was utilized, while Pandas facilitated the data manipulation tasks.

### 4.1. Dataset Description

We describe the datasets obtained from sensors in real-world scenarios to monitor water quality is depicted in Table 1. These datasets serve as the foundation for evaluating the performance of our suggested anomaly detection system. The supplied dataset contains data from the USGS 06041000 Madison River near Ennis Lake station, which is located in McAllister, Montana. The file contains several parameters and statistical measures about the water conditions at this station. The following is a description of the available data:

Height in feet (mean)—parameter code: 00065, statistic code: 00003: This measure denotes the average gauge height or water level at the station in feet. Gauge height is a measurement of the water level wat a fixed location.

Water temperature (maximum) in degrees Celsius—parameter code: 00010, statistic code: 00001. This shows the highest recorded water temperature in degrees Celsius. It tells you the hottest temperature ever recorded in the water.

Water temperature in degrees Celsius (minimum)—parameter code: 00010, statistic code: 00002. The minimum recorded water temperature in degrees Celsius is represented by this measure. It provides information on the water’s lowest temperature.

Water temperature in degrees Celsius (mean)—parameter code: 00010, statistic code: 00003. In degrees Celsius, this Figure represents the average or mean water temperature. It gives information about the average water temperature.

Mean discharge in cubic feet per second—parameter code: 00060, statistic code: 00003. The average discharge or flow rate of water in cubic feet per second is represented by this measurement. It provides information about the volume of water moving through the station.

The dataset may also include data-value qualification codes, indicating the data’s quality or dependability. The file contains qualification codes such as “A” for approved data, “P” for provisional data subject to amendment, and “e” for estimated values. Multiple entries exist in the dataset, each with corresponding values for different parameter/statistic combinations, representing separate time series instances (measurement intervals).

Furthermore, we examine a multivariate time series model for forecasting water quality across time to recreate the primary characteristics of water quality changes. The model uses a collection of time series measurements to examine the temporal persistence of water quality indicators such as turbidity (TURB), specific conductance (SC), and dissolved oxygen (DO).

The information offered comprises measurements collected from sensors to monitor water quality.

The dataset encompasses many metrics, including gauge height, maximum, minimum, mean water temperature, and mean discharge, among other variables.

In order to establish the accuracy and reliability of the measurements obtained from the designed sensor in comparison to those obtained from the commercial sensor, the following steps were undertaken:

Parallel measurements: Simultaneously perform measurements using both sensors under identical environmental circumstances. This practice guarantees that both sensors are subjected to identical variables that have the potential to impact the recorded data.

Statistical analysis: Conduct statistical studies on the measurements acquired from both sensors. In order to assess the degree of concordance among the sensors, it is necessary to compute statistical measures, including the mean.

In the first stage of the data training process, sub-samples of the training dataset, as shown in Figure 3, are used. Then, using the MCN-LSTM, an anomaly score is calculated for each test case.

### 4.2. Experimental and Discussion

In this section, we conduct experiments on the suggested models using publicly available Internet datasets. We contrast these models with other deep-learning models created primarily for time-series anomaly identification. All the experiments described in this section were conducted using a computing setup equipped with an Intel(R) Core(TM) i5-8250U CPU, a GeForce MX150 GPU, and 8 GB of RAM. Python served as the primary programming language, and PyTorch’s deep learning framework was employed. For numerical calculations, NumPy was utilized, while Pandas facilitated the data manipulation tasks. Additionally, Google Colab was utilized as part of the experimental environment.

The experimental design, including the evaluation metrics and parameters used in the MCN-LSTM model, is detailed in this section. We discuss the rationale behind the chosen metrics and how they relate to the specific challenges of water quality anomaly detection.

We present the experimental results of the MCN-LSTM technique applied to the water quality datasets. MCN-LSTM was tested on a water quality dataset, and the best number of LSTM cells for the dataset was consistently chosen to be 128, while the hyperparameters remained constant with kernel size. As proposed by Hu et al. (2017) [32], we use 16 as the reduction ratio, r, for all the squeeze-and-excitation blocks. To avoid overfitting, we set the total number of training epochs to 250 and the dropout rate to 80% throughout the training phase. Each of the proposed models is trained with 128 batches. The uniform He initialization approach described by He, Zhang, Ren, and Sun (2015) [33] is used to initialize the convolution kernels. To exhibit anomaly detection rate and false alarm/fall-out rate, two single measures, true positive rate (TPR) and false positive rate (FPR) are utilized as a one-dimensional metric approach [34]. True positive indicates that actual anomalous observations have been detected as anomaly conditions using the anomaly detection method. A false positive is when actual normal observations are identified as anomalous conditions by an anomaly detection algorithm. TPR and FPR can be computed as follows (Perelman et al., 2012) [34]. The kernel parameter of the SVM model has the potential to impact the generalization capability inside the feature space, leading to the occurrence of overfitting or underfitting phenomena [35]. Hence, a systematic trial-and-error methodology was employed to ascertain the optimal σ value, which spanned the range of 0.1 to 10.0. The increase for the σ value ranges of 0.1–1.0 and 1.0–10.0 was 0.1 and 1.0, respectively.
(5)$$TPR=\frac{TP}{TP+FN}=Sensitivity$$
(6)$$FPR=\frac{FP}{TN+FP}=1-Specificity$$

This means the real water quality is normal, as is the anomaly detection result.

The created anomaly detection approach is analyzed and validated using a time series of water quality measures such as TURB and SC.

The MCN-LSTM approach has been used in the experiment to detect anomaly samples for dissolved oxygen data as shown in Figure 4. Figure 5 shows the position of all the anomaly instances in the test dataset.

Figure 5 shows the result of the experiment on the water quality dataset. The MCN-LSTM approach has been used in the experiment to detect anomaly samples for dissolved oxygen data. Figure 5 shows the position of all the anomaly instances in the test dataset.

### 4.3. Comparison with Other Approaches

We compare the performance of our proposed method with existing anomaly detection techniques for water quality data. We discuss each method’s strengths and weaknesses and highlight the MCN-LSTM approach’s advantages.

Real-world application and case studies: To showcase the practical utility of our proposed approach, we present real-world case studies where the Multivariate deep learning technique effectively identified water quality anomalies in real time.

In the experiment, the MCN-LSTM technique was employed to detect anomalies in temperature data. Figure 6 depicts the location of each anomaly incidence in the test dataset.

In Figure 6, the technique was trained on 1000 random train–test splits for different numbers of train samples. We use the MCN-LSTM technique for the anomalous water quality dataset and the root mean squared error for the sample of the water quality dataset.

The training and validation functions for the LSTM model with layers and the MCN-LSTM technique are illustrated in Figure 6. On the x-axis, the number of time series is depicted, while the y-axis represents temperature values. Examining the elbow points in the training values, which denote the point where an increase in the number of time series results in only a marginal difference in the measured values, it becomes evident that MCN-LSTM achieves a stable model more rapidly with less error in fewer epochs. This observation is reasonable, given that the convolutional layers in MCN-LSTM enhance the learning process in each epoch.

We concentrated on improving the F1 score because it is the best indicator for evaluating our experiment and is also utilized by MCN-LSTM to select the best model. The F1 score of each trained model is depicted in Table 2.

This section aims to analyze and compare the MCN-LSTM (Multivariate Convolutional Neural Network—Long Short-Term Memory) technique and the LSTM (Long Short-Term Memory) technique in the context of anomaly identification utilizing water quality sensor data. The MCN-LSTM model demonstrates a smaller mean absolute error (MAE), which suggests a higher accuracy level in forecasting average water quality data. A decreased mean absolute error (MAE) is considered advantageous in anomaly detection since it indicates a reduced absolute disparity between the anticipated and observed values.

The Long Short-Term Memory (LSTM) model has a reduced mean squared error (MSE), indicating a tendency towards less squared errors on average. Nevertheless, the disparity is not significant. The mean squared error (MSE) metric quantifies the dispersion of errors, and within the domain of anomaly detection, both models exhibit a satisfactory level of performance.

The Long Short-Term Memory (LSTM) model has a reduced root mean square error (RMSE), which suggests a diminished average magnitude of errors. This observation implies that LSTM might exhibit a higher level of precision in identifying abnormalities, given the quadratic nature of the RMSE metric.

In the context of the entire assessment, as shown in Figure 7, the MCN-LSTM model demonstrates a lower mean absolute error (MAE), suggesting greater accuracy. The MCN-LSTM exhibits a somewhat elevated mean squared error (MSE) and root mean squared error (RMSE) compared to the LSTM. This individually demonstrates a high proficiency level in accurately detecting water quality anomalies.

The Long Short-Term Memory (LSTM) model exhibits a mean absolute error (MAE) that is reduced compared to the MCN-LSTM model, albeit still greater than the intended threshold. The performance of the water quality sensors exhibits a minor improvement in terms of the mean squared error (MSE) and root mean squared error (RMSE), indicating enhanced precision in detecting abnormalities. This observation should be considered when considering the suitability of these sensors for water quality monitoring purposes.

The multivariate technique employed by MCN-LSTM can offer advantages when water quality data consist of numerous variables. This approach allows complicated correlations to be captured, leading to a more accurate detection of anomalies.

The Long Short-Term Memory (LSTM) model’s capacity to capture extended temporal dependencies should prove advantageous in analyzing water quality data that exhibit complex temporal patterns.

MCN-LSTM may be considered a favorable option when dealing with multivariate water quality data exhibiting intricate interactions.

If the temporal dynamics of the data play a vital role and a little less complex model is deemed acceptable, Long Short-Term Memory (LSTM) could be a viable alternative.

The present analysis and comparison offer valuable insights into the respective merits and drawbacks of the MCN-LSTM and LSTM models in the context of water quality anomaly detection, taking into account the performance metrics used. The selection between the two options is contingent upon the unique features of the water quality data and the intended compromises between precision and model intricacy.

## 5. Conclusions and Future Work

The Multiple Convolutional Networks and Long Short-Term Memory (MCN-LSTM) technique was used in this paper to present a unique approach for real-time anomaly detection in water quality data. The growing use of automated systems, the Internet of Things (IoT), and sensors for water quality monitoring necessitates the early detection of unexpected values. Anomalies in water quality data can seriously affect decision making and environmental protection. Our MCN-LSTM technique displayed outstanding accuracy in detecting anomalies, with an impressive 92.3% accuracy in differentiating between normal and anomalous data instances. An extensive experimental study of real-world water quality measurements acquired from sensors proved our approach’s success. The quantitative results support the ability of MCN-LSTM to improve decision-making processes and avoid negative repercussions caused by undetected abnormalities. Our technique contributes to the safety and sustainability of water resources by detecting anomalies in real time, helping both public health and environmental conservation.

Future work: While our proposed MCN-LSTM technique has yielded promising results, there are a number of avenues for additional investigation and development. To begin, we can study the impact of changing hyperparameters and model topologies on deep learning performance. Fine-tuning the model may improve its anomaly detection skills. Additionally, integrating datasets from diverse geographic regions and seasons could improve the approach’s resilience. Furthermore, combining the MCN-LSTM technique with real-time data streaming and IoT technologies could enable continuous monitoring and rapid response to anomalies, boosting the accuracy of water quality evaluation even further.

Furthermore, experimenting with ensemble learning approaches and merging various anomaly detection models could significantly improve detection accuracy while reducing false positives. Conducting comparative research with other cutting-edge anomaly detection algorithms, including classic machine learning methods, would provide a thorough assessment of the strengths and drawbacks of various approaches for water quality monitoring. Finally, the suggested MCN-LSTM approach has shown considerable promise in detecting water quality anomalies in real time. We can progress the state of water quality monitoring by refining the model and investigating possible extensions, as well as contribute to the sustainable management of water resources, ensuring their safety and preservation for future generations.

## Figures and Tables

**Figure 1 sensors-23-08613-f001:**
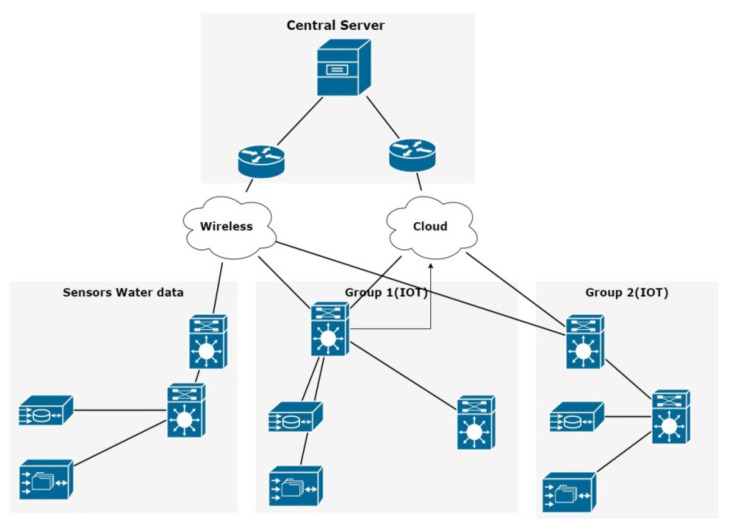
Structure of water monitoring system.

**Figure 2 sensors-23-08613-f002:**
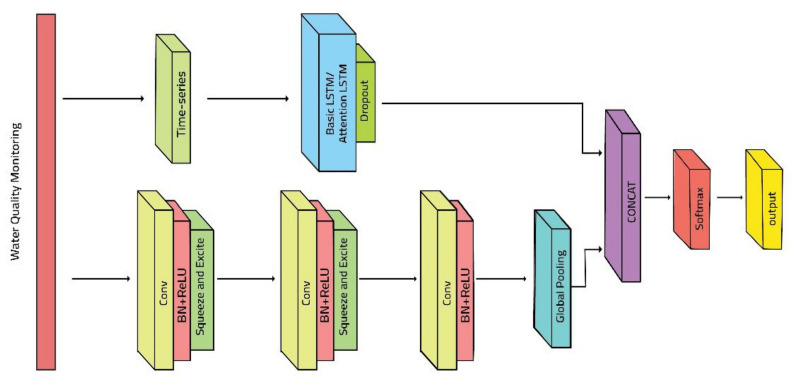
Structure of MCN-LSTM for real-time anomaly detection.

**Figure 3 sensors-23-08613-f003:**
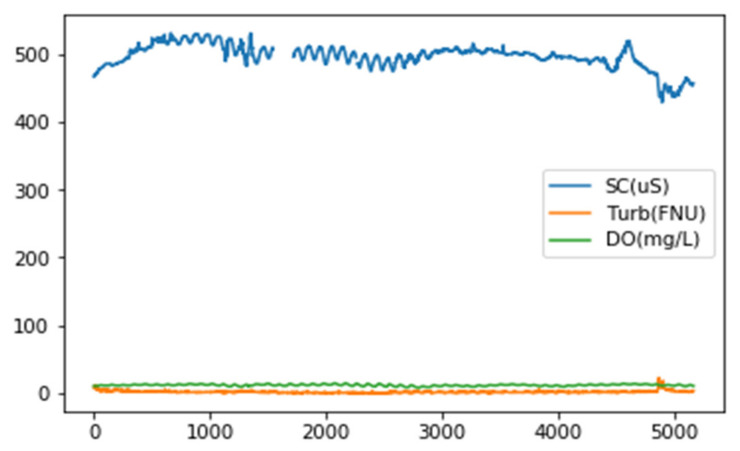
Sample of measurements from Real-time water quality dataset.

**Figure 4 sensors-23-08613-f004:**
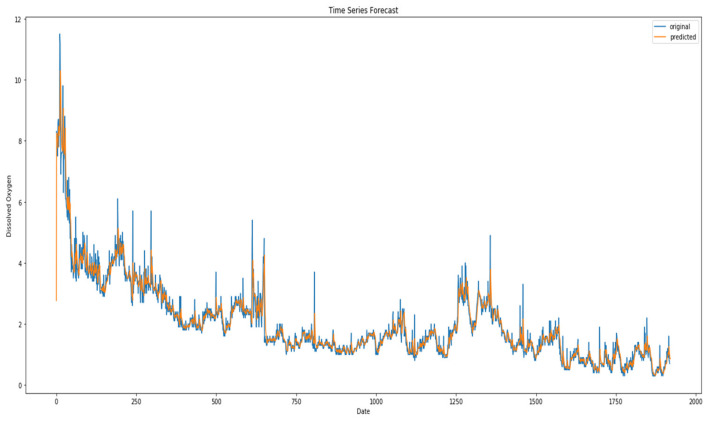
Sample of anomalies for dissolved oxygen in water quality dataset using MCN-LSTM technique.

**Figure 5 sensors-23-08613-f005:**
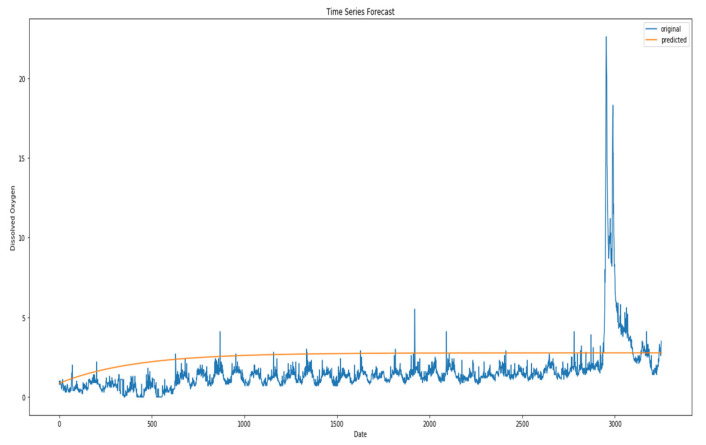
Sample of anomalies for dissolved oxygen in water quality dataset.

**Figure 6 sensors-23-08613-f006:**
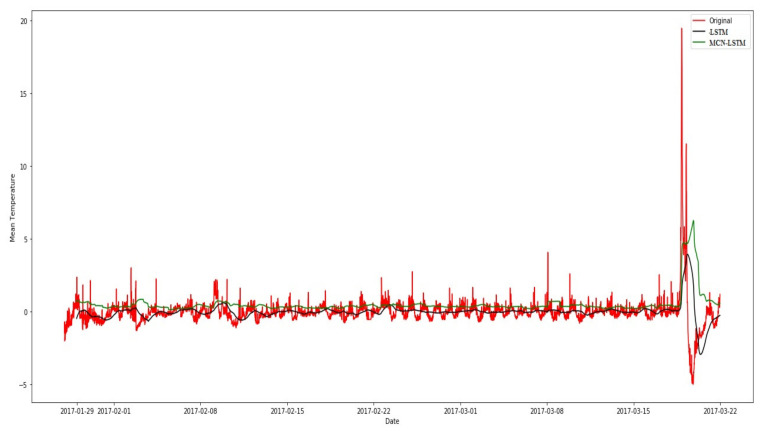
Comparison between the MCN-LSTM technique and the other techniques.

**Figure 7 sensors-23-08613-f007:**
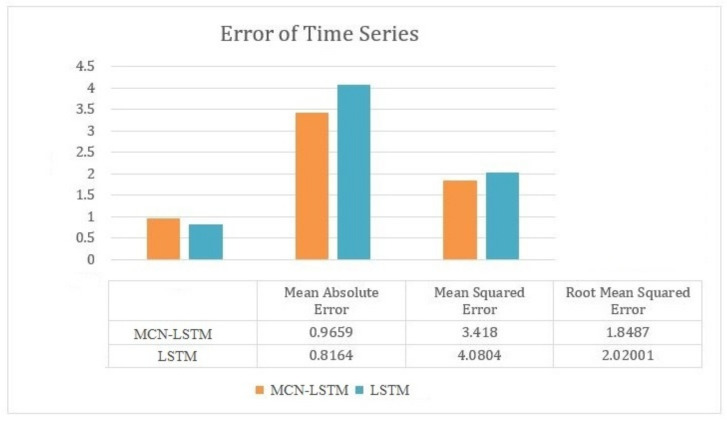
Comparison between the MCN-LSTM technique and the LSTM techniques.

**Table 1 sensors-23-08613-t001:** Sample of measurements from sample dataset.

Date	Time	SC (μs)	Turb (FNU)	DO (mg/L)
27 January 2017	00:00:00	467.0	8.3	10.4
	00:15:00	467.0	NaN	10.4
	00:30:00	467.0	7.8	10.4
	00:45:00	467.0	8.0	10.4
	01:00:00	468.0	7.5	10.4
	01:15:00	468.0	8.6	10.4
	01:30:00	469.0	8.7	10.4
	01:45:00	468.0	NaN	10.4
	02:00:00	469.0	7.8	10.4
	02:15:00	469.0	8.5	10.4
	02:30:00	469.0	NaN	10.4
	02:45:00	469.0	11.5	10.4
	03:00:00	470.0	11.2	10.4
	03:15:00	470.0	9.1	10.4
	03:30:00	470.0	6.9	10.4
	03:45:00	470.0	7.4	10.4
	04:00:00	470.0	7.6	10.4
	04:15:00	471.0	NaN	10.4
	04:30:00	471.0	NaN	10.4
	04:45:00	471.0	7.7	10.4
	05:00:00	471.0	9.8	10.4
	05:15:00	472.0	NaN	10.5
	05:30:00	472.0	6.3	10.5
	05:45:00	472.0	7.4	10.5
	06:00:00	473.0	NaN	10.5
	06:15:00	472.0	8.0	10.5
	06:30:00	473.0	8.8	10.5
	06:45:00	473.0	NaN	10.5
	07:00:00	473.0	6.3	10.5
	07:15:00	474.0	6.1	10.5

NaN refere to (Data is Missing).

**Table 2 sensors-23-08613-t002:** Technique’s performance using time series.

Techniques	F1	FPR	TPR
SVM	0.82	0.99	0.40
RNN	084	1.0	0.12
LSTM	0.88	0.04	0.86
CNN	0.92	0.93	0.49
ANN	0.87	0.11	0.85
MCN-LSTM	0.93	0.20	0.97

## Data Availability

The data presented in this paper are available on request.

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
