# Peer review of "Real-Time Anomaly Detection for Water Quality Sensor Monitoring Based on Multivariate Deep Learning Technique"

_sensors, 2023, doi:10.3390/s23208613_

Round 1
Reviewer 1 Report (Previous Reviewer 2)
In this paper, a combination of multiple convolutional networks and long short-term memory (MCN-LSTM) technique is designed to effectively identify anomalies in time series data. There are some problems in the paper, which I suggest to reconsider after considering the following suggestions. Some of the questions that arise are listed below:
1. In the introduction and related work part, little references on real-time anomaly detection using deep learning technology in water quality monitoring published in recent three years were referred, and some research results can be summarized to supplement this part.
2. This article combines several techniques, but this approach can be applied to other areas as well. I suggest the author list how this paper contributes to current research? and what new knowledge is added by this study?
3. There are many methods of real-time anomaly detection using deep learning technology in water quality monitoring, and it is suggested that the author can summarize the limitations of these methods and propose ways to solve these limitations, which can reflect the novelty of the new technology.
4. Line 356-360, Line 386: Formulas need to be numbered, generally at the end of the right line.
5. Line 337-396: This part of the content is relatively weak in the manuscript, you need to focus on it to further strengthen the connections between paragraphs and show technical highlights and details.
6. Line 449-480: In the experimental part, the detailed description of the experimental process is not enough. It is suggested that the author can describe the experimental process clearly, which is conducive to proving the authenticity of the proposed method.
7. Line 481-494: In the comparison experiment part, this method should be compared with the current more advanced water quality monitoring methods. In Figure 6, it is recommended that the author clearly describe how this method is superior to other methods, because the diagram is not intuitive enough.
Author Response
"Please see the attachment."

Reviewer 2 Report (New Reviewer)
Comments to authors:
The study is interesting with promising results. However, the structure of the paper requires huge revision. For instance:
(i) Section 3: The proposal MCN-LSTM technique
It should be 3. Materials and Methodology
3.1 Proposed MCN-LSTM Technique
(ii) Same happens to section 4: It should be 3.x for the subtopic, but not standing as a new subtopic.
(iii) Section 5 Should be 4. Results and Discussions. And write properly what is the Experimental. More precise description should be provided.
(iv) The introduction of the study needs to be revised. Clear research objectives and gaps should be mentioned. Not clear at the moment. Some relevant references should be included:
(a) Toward industrial revolution 4.0: Development, validation, and application of 3D-printed IoT-based water quality monitoring system
(v) The discussion section is lacking in this study. You should try to discuss more relevant reasonings about the ML methods you chose and reasons why one outperformed another. You may refer to Application of artificial intelligence methods for monsoonal river classification in Selangor river basin, Malaysia
(vi) You should be providing a flow chart of the study. It is not too clear.
(vii) Also give us the schematic diagram of your water quality sensor.
(viii) How was the validation or verification of measurements between the developed sensor and the commercial sensor? How was the fittings between these two sensors? Good or poor?
(ix) What is your evaluation criteria? Lots of discussion is needed.
Moderate editing of English language required
Author Response
"Please see the attachment."

Round 2
Reviewer 1 Report (Previous Reviewer 2)
In this paper, a combination of multiple convolutional networks and long short-term memory (MCN-LSTM) technique is designed to effectively identify anomalies in time series data. Some of the questions that arise are listed below:
1. Line 194-219: In the introduction, a variety of techniques for abnormal water quality monitoring are mentioned. It is suggested that the author describe these methods according to the historical development process and summarize the existing shortcomings of each type of technology in one sentence, so as to propose MCN-LSTM technology.
2. Line 16-34: The flow chart of the study should include five parts: data collection, feature extraction, water quality prediction, anomaly monitoring and result analysis. It is suggested that the author should make the chart more beautiful.
3. You only used one evaluation criterion. It is suggested that the author adopt multiple evaluation indexes, such as RMSE, MAE, MAPE, etc., to improve the explanatory power of the experimental results.
4.It is suggested that the author should pay attention to the cohesive relationship between each part of the content and each paragraph, and the description of the content should be logical and promote this relationship.
Author Response
Please see the attachmen

Reviewer 2 Report (New Reviewer)
The authors have substantially addressed my comments during the first review. For the section 1.3 Industry 4.0 for 3D printing in sensor water quality. The naming of the section sounds weird. It is suggested to modify to Industry 4.0 for 3D printed water quality sensor.
Also, cite the relevant reference as suggested in the first review: Toward industrial revolution 4.0: Development, validation, and application of 3D-printed IoT-based water quality monitoring system.
Again, the result and discussion section is very short in this study. A good manuscript should be having a good balance across different sections. The discussion section is lacking in this study. You should try to discuss more relevant reasonings about the ML methods you chose and reasons why one outperformed another. Please cite Application of artificial intelligence methods for monsoonal river classification in Selangor river basin, Malaysia in your revised manuscript.
Revision is required.
Moderate editing of English language required
Round 3
Reviewer 2 Report (New Reviewer)
Authors have substantially addressed my comments.
Minor editing of English language required
This manuscript is a resubmission of an earlier submission. The following is a list of the peer review reports and author responses from that submission.
Round 1
Reviewer 1 Report
1. The given introduction and related work can be improved to be more focused.
2. The authors need to highlight more contemporary works that use Multiple Convolutional Networks in the related works section.
3. To achieve novelty using new techniques, it is important to identify the limitations of existing methods and propose solutions that address those limitations.
4. You need to clearly define what pieces of data are required to work the model and how this data relates to the equations you have provided.
5. The paper combines many techniques. I mean, it seems that the method can be applied to other tasks. Thus, it is hard for me to discern the contributions.
6. The experimental section should be substantially improved. Such a method should be compared with recent methods on water quality monitoring problems. Additionally, conducting thorough experimental evaluations and providing comparative analysis with existing methods can demonstrate the uniqueness and effectiveness of the proposed approach.
Reviewer 2 Report
In this paper, a combination of multiple convolutional networks and long short-term memory (MCN-LSTM) technique is designed to effectively identify anomalies in time series data. We offer the following suggestions for authors to modify.
Point 1#
The abstract of the paper does not conform to the norms of computer paper writing, and the core technology is only briefly mentioned, without a good description.
Point 2#
The reference [3] in section 1.1 should not be here, but at the end of the sentence.
Point 3#
In the introduction, the author does not describe the problems and shortcomings of traditional technology and machine learning technology. At the same time, the authors do not describe how the proposed deep learning model is good, and which problems and shortcomings of existing technologies are solved.
Point 4#
The author neglects many important studies related to the research status of anomaly detection model of water quality data. In related work, the authors apparently do not have enough information about existing studies and suggest that the authors must examine more studies done by others. At the same time, it is recommended that the author write this part of the content in segments, a whole paragraph of content will give the reader a bad reading experience.
Point 5#
The drawing of Figure 2 in Chapter 3 is not beautiful, and the formula is not standardized, there is no formula number, there is no specific description for each symbol in the formula, and the description of the formula needs to be more detailed. At the same time, there is no clear description of the advantages of the core technology, and there is no connection between paragraphs, which needs to pay attention to the logic of the language.
Point 6#
The dataset is not well described in Chapter 4 and needs to be more detailed.
Point 7#
Experimental environment Can CPU i5-8250U, GPU MX150, and 8GB storage support this experiment and run a large number of data sets? At the same time, the experimental process does not meet the requirements and requires data processing, small batches of experiments, training times, dropout probability between layers, time step, number of neurons and learning rate, etc., so the author needs to be more detailed.
Point 8#
The comparative experiments are not enough to demonstrate the validity of the proposed model, and the relevant ablation experimental process is lacking! The author needs to compare with some models and reflect the advantages of the proposed model after detailed analysis.
Point 9#
It is suggested that the author summarize the work completed in this paper.
Point 10#
The novelty and contribution of the paper need to be further strengthened.
Point 11#
It is recommended that authors cite literature within the last 5 years as much as possible.